# LightCache: Memory-Efficient, Training-Free Acceleration for Video Generation

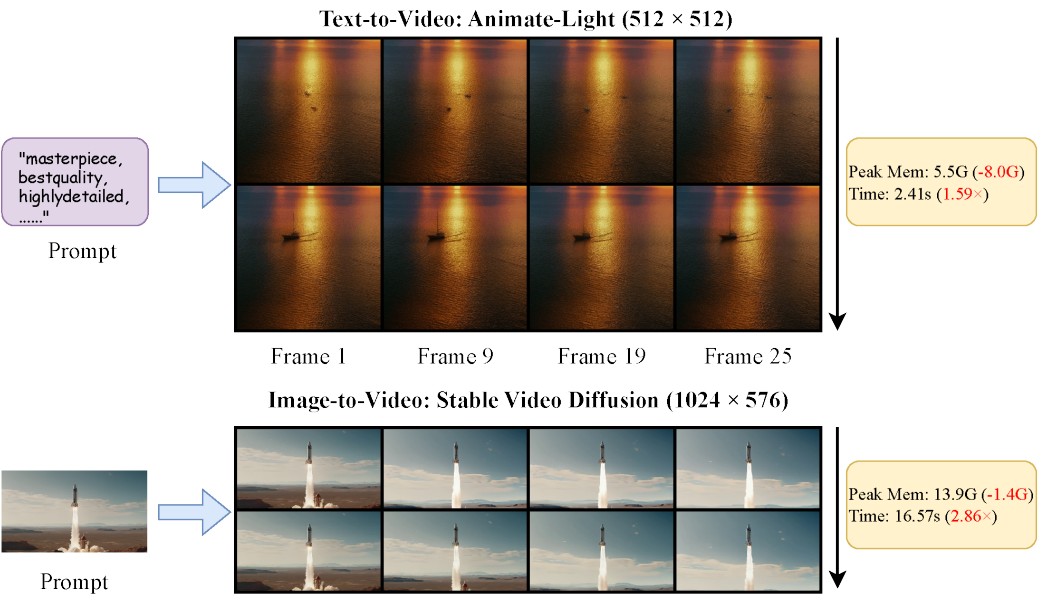

Figure 1: Accelerating AnimateDiff-Lightning and Stable-Video-Diffusion-Img2vid-XT by 1.59× and 2.86×, while reducing memory usage by 8.0 GB and 1.4 GB, respectively.

## Abstract

Training-free acceleration has emerged as an advanced research area in video generation based on diffusion models. The redundancy of latents in diffusion model inference provides a natural entry point for acceleration. In this paper, we decompose the inference process into the encoding, denoising, and decoding stages, and observe that cache-based acceleration methods often lead to substantial memory surges in the latter two stages. To address this problem, we analyze the characteristics of inference across different stages and propose stage-specific strategies for reducing memory consumption: 1) Asynchronous Cache Swapping. 2) Feature chunk. 3) Slicing latents to decode. At the same time, we ensure that the time overhead introduced by these three strategies remains lower than the acceleration gains themselves. Compared with the baseline, our approach achieves faster inference speed and lower memory usage, while maintaining quality degradation within an acceptable range. The Code is available at `https://anonymous.4open.science/r/LightCache-CD86`.

## 1 Introduction

Diffusion models have recently received significant global attention due to their impressive generative capabilities Croitoru et al. (2023); Yang et al. (2023); Rombach et al. (2022). These models have demonstrated remarkable efficacy in various applications, like the generation of images Peebles & Xie (2023); Ho et al. (2020); Song et al. (2020), language Hoogeboom et al. (2021); Austin et al. (2021); Yang et al. (2025), and video Blattmann et al. (2023); Guo et al. (2023); Yang et al.

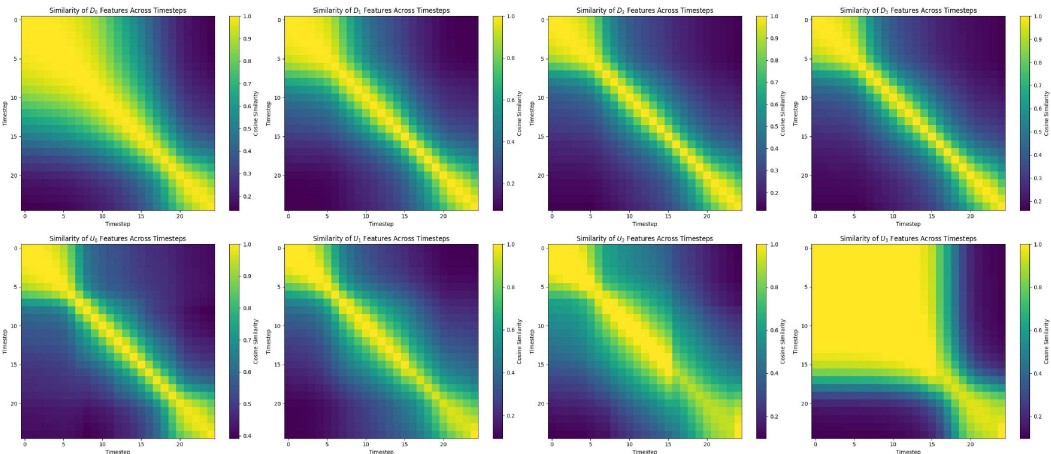

Figure 2: Highly similar feature map between all up-sampling and down-sampling layers

(2024). However, due to the problems of high computational time cost and large memory usage, researchers are often seriously restricted to their implementation and deployment in the real world Ma et al. (2024a). Especially in the video generation task process, the model must simultaneously perform a denoising action on all frames, leading to a high GPU memory usage and a long inference latency Xing et al. (2024). To solve this problem, existing studies have attempted to achieve acceleration through training Luo et al. (2023); Lyu et al. (2022); Yin et al. (2024); Wang et al. (2023); Kim et al. (2023); Ma et al. (2024a); Wimbauer et al. (2024); Agarwal et al. (2024) and training-free methods Ma et al. (2024b); Zhan et al. (2024); Yu et al. (2023); Li et al. (2023b;a).

Training methods require a lot of training resources and may have a negative impact on the generation performance of the original model Li et al. (2023c); Ma et al. (2024b); Yang et al. (2023); Ma et al. (2024a); Zhou et al. (2025). In contrast, training-free methods for acceleration have focused on two main strategies Ma et al. (2024a); Liu et al. (2025): 1) skip certain time steps in the inference stage Ma et al. (2024b), and 2) decrease the inference overhead per step through methods Zhan et al. (2024); Yu et al. (2023); Li et al. (2023b). The inherent redundancy in the feature maps during the denoising stage provides a natural entry point for skipping certain inference steps Li

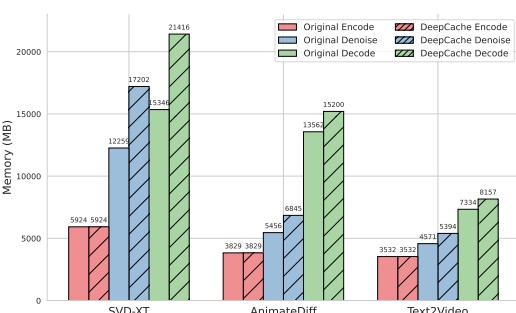

Figure 3: Memory Usage

et al. (2023a). Based on our experimental observations and a review of existing literature Ma et al. (2024b); Li et al. (2023a); Tian et al. (2025); Li et al. (2023b; 2025), we found that adjacent denoising steps often produce highly similar intermediate representations (See in Figure 2), particularly in the later stages of inference when the signal becomes clearer and most of the noise has been eliminated. This temporal coherence suggests that many feature maps encode overlapping semantic information, thereby enabling the possibility of reducing the number of inference steps without significantly compromising the quality of the generated results.

Our research is based on a cache mechanism that skips the computation of certain feature maps at specific timesteps to accelerate video generation. However, the existence of caching directly leads to a surge in GPU memory usage (See in Figure 3). For example, when we applied DeepCache to EasyAnimate (a DiT architecture) using 4 L40 GPUs (45 GB each), we immediately encountered an out-of-memory issue. This happens because model inference differs from model training. During training, the common practice is either data parallelism or model parallelism (tensor/layer parallelism). In data parallelism, a full copy of the model is placed on each GPU, while the sample batch is evenly divided, leading to relatively balanced memory usage. In model parallelism, the

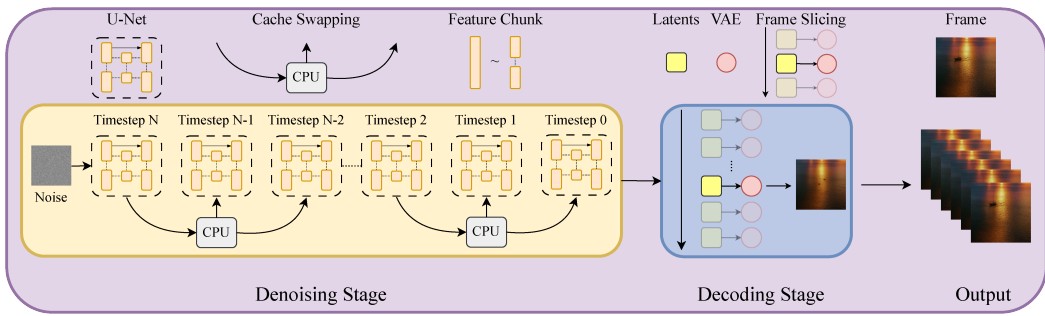

Figure 4: **Denoising:** The timestep is divided into a cache step and a normal step. The cache step reuses the cached feature maps stored in the CPU. The normal step chunk the size of feature maps and concatenate them. **Decoding:** We use VAE to decode the latent frame by frame.

model is split across GPUs, so memory is also distributed more evenly. However, during model inference/sampling, if we only generate a single sample (batch = 1), there is no natural way to distribute the workload as in training. Multi-GPU parallelism can only assign different modules to different GPUs, but the memory usage of each module is not the same. If a single module (e.g., U-Net or DiT) already exceeds the capacity of a single GPU, then even with multi-GPU parallelism, out-of-memory errors will occur.

Therefore, we also hope to reduce the GPU burden of the inference stage. To explore the sources of computation and memory cost, we divide the inference into three stages: **encoding**, **denoising**, and **decoding** stages, and measured the peak memory(See in Figure 3). That is, the memory requirement for diffusion inference depends on the highest memory of the three stages. For example, when we applied the method Zhan et al. (2024) to reduce memory usage, we found that the peak memory did not decrease (We use the default setting in the diffusers library). According to our experiments, this method focuses on the denoising stage (See in Tab. 2), while the peak memory of some models or some experimental settings is in the decoding stage(See in Figure 3). Empirically, regardless of whether DeepCache Ma et al. (2024b) is used for accelerating inference, the memory usage of the encoding stage remains consistently low and unchanged. Therefore, memory optimization efforts should focus on the denoising and decoding stages.

We propose the following optimization strategy (See in Figure 4):

- **Chunk**: The denoising stage primarily involves the computation of feature maps. We propose that selectively chunking the height and width of feature maps from different layers to varying degrees can effectively reduce memory usage while maintaining generation quality.

- **Asynchronous Swapping**: The model offloading function von Platen et al. (2022) in the Diffusers library moves inactive layers or models to the CPU to reduce GPU memory usage. Following a similar idea, swapping inactive cached feature maps to the CPU could also alleviate memory pressure. To further mitigate the latency overhead, asynchronous swapping can be employed: feature maps can be transferred between GPU and CPU in the background while the GPU continues computing other parts of the model, thereby overlapping communication and computation.

- **Slicing**: The decoding stage can be optimized by splitting large inputs into smaller batches and processing them sequentially. In our case, although we generate only a single video, the generative models we adopt are classifier-free guidance models Ho & Salimans (2022). This naturally doubles the number of feature maps (conditional and unconditional) and requires denoising across multiple frames simultaneously (a total of $2 \times N$, where $N$ is the number of frames). Thus, VAE slicing remains highly applicable.

## 2 METHODOLOGY

### 2.1 PRELIMINARY

**Diffusion Forward Process.** The core idea of diffusion models Ho et al. (2020) is to remove noise from fully corrupted data in order to recover valid semantic information using a series of timesteps. Its essence is to learn the transformation of noise distribution to ground truth distribution. For the data $\mathbf{x}$, we can gradually add Gaussian noise to the data in T steps:

$$q(\mathbf{x}_t \mid \mathbf{x}_{t-1}) = \mathcal{N}(\mathbf{x}_t; \sqrt{1 - \beta_t}\mathbf{x}_{t-1}, \beta_t\mathbf{I}) \tag{1}$$

where $t$ belongs to $[0, T]$, and $[\beta_0, ..., \beta_T]$ schedule the noise.

**Diffusion Reverse Process.** To recover the valid semantic information, we train the diffusion model to denoise the randomly sampled noise into the ground truth distribution. We use a network (For example, U-Net Ronneberger et al. (2015)) $\epsilon(\mathbf{x}_t, t)$ to predict noise:

$$p_\theta(\mathbf{x}_{t-1} \mid \mathbf{x}_t) = \mathcal{N}(\mathbf{x}_{t-1}; \frac{1}{\sqrt{\alpha_t}}(\mathbf{x}_t - \frac{1 - \alpha_t}{\sqrt{1 - \bar{\alpha}_t}}\epsilon_\theta(\mathbf{x}_t, t)), \beta_t\mathbf{I}) \tag{2}$$

where $\alpha_t = 1 - \beta_t$ and $\bar{\alpha}_t = \prod_{i=1}^T \alpha_i$. Applied iteratively, it gradually removes the noise of the current $\mathbf{x}_t$, bringing it close to a real data point when we reach $\mathbf{x}_0$.

**Feature Calculation** U-Net Ronneberger et al. (2015), originally proposed for medical image segmentation, uses skip connections to effectively combine low- and high-level features. Its architecture is built on successive down-sampling and up-sampling blocks that first encode the input into a compact representation and then decode it for downstream applications:

$$\{D_i\}_{i=0}^m, \{U_i\}_{i=0}^m \tag{3}$$

Down-sampling is the encoding stage, which extracts the information from the previous layer through the $\{D_i\}_{i=0}^m$. Up-sampling is the decoding stage, which converges the inversion from the previous layer and the skip-connection:

$$U_i = \text{Concat}(D_i, U_{i+1}) \tag{4}$$

Therefore, at the heart of a U-Net model is a concatenation of low-level features from the skip connection and the high-level features from the previous layer.

### 2.2 ASYNCHRONOUS SWAPPING

We introduce DeepCache Ma et al. (2024b), a simple yet effective method that exploits temporal redundancy between steps in the diffusion reverse process to accelerate inference, serving as the foundation of our approach. As the cacheing mechanism Smith (1982) in a computer system and KVCache in transformer Pope et al. (2023), DeepCache method caches the slowly evolving features $F^t$ to eliminate some unnecessary computations in some steps:

$$\begin{aligned} F_{cache}^t &\leftarrow D_m^t, \text{Concat}(D_m^t, U_{m+1}^t) \\ D_m^{t+1}, U_m^{t+1} &\leftarrow F_{cache}^t \end{aligned} \tag{5}$$

Where $U_m^t$ or $D_m^t$ means the $m$-th up-sampling layer or down-sampling layer at time step $t$.

For example, in the case of an up-sampling layer, we set a hyperparameter $n$, meaning that the feature map is explicitly recalculated once every $n$ steps, while in the remaining steps it is reused from the cache: $F_{cache}^t : U_m^0, U_m^n, U_m^{2n}, \ldots$ There is no additional computational cost to these time steps: $\{(1, \ldots, n-1); (n+1, \ldots, 2n-1); \ldots\}$, since it can be simply retrieved from the cached feature map. Although DeepCache enables feature map reuse for acceleration, the additional cached feature maps lead to a substantial increase in memory usage.

In the official diffusers library, CPU offloading von Platen et al. (2022) selectively transfers weights between the GPU and CPU. When a component is needed, it is moved to the GPU, and when it is

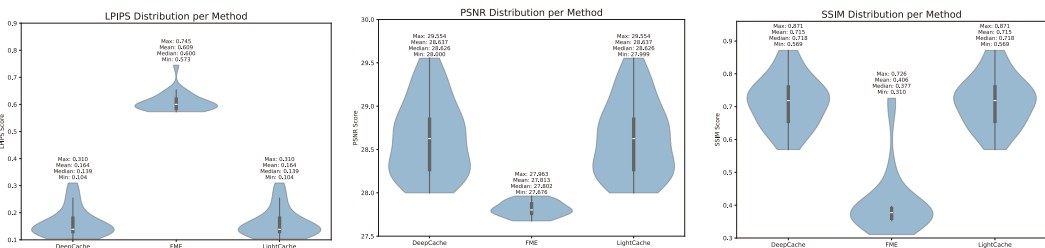

Figure 5: Violin Figure (AnimateDiff-Light) of Quality Metric: LPIPS($\downarrow$), PSNR($\uparrow$), SSIM($\uparrow$)

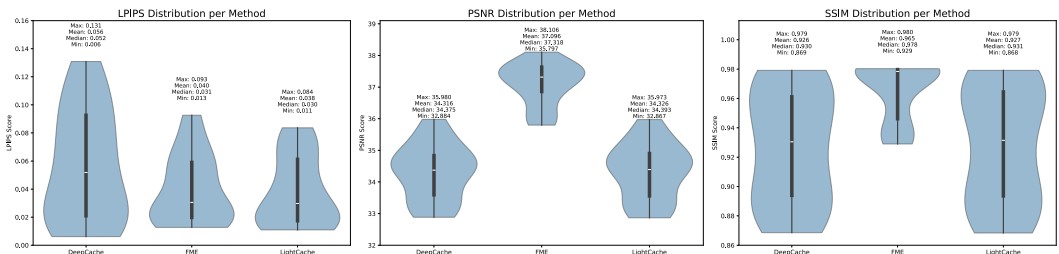

Figure 6: Violin Figure (SVD) of Quality Metric: LPIPS($\downarrow$), PSNR($\uparrow$), SSIM($\uparrow$)

not needed, it is offloaded back to the CPU. This method operates on submodules rather than entire models, thereby saving memory by avoiding storing the whole model on the GPU. Inspired by this idea, we store the cached features on the CPU and bring them back when required. This approach does not affect generation quality, though it slightly increases inference time, while significantly reducing GPU memory usage.

## 2.3 FEATURE CHUNK AND SLICED DECODING

So far, we have not modified the fundamental feature process of the diffusion model. The input $F_m^t \in \mathbb{R}^{B \times T \times C \times H \times W}$ of (m+1)-th layers is a 5-D feature map with dimensions {batch, frames, channels, height, width}, where the batch size is 2, corresponding to the conditional and unconditional feature maps in the classifier-free guidance model. Zhan et al. (2024) found that directly modifying the batch or frame dimensions would break the spatiotemporal continuity of video generation, leading to distorted temporal information extraction. On the other hand, directly modifying the channel dimension would cause misalignment with the model architecture. Therefore, they keep the first three dimensions unchanged and apply chunking only to the spatial dimensions (height and width):

$$Chunk(F_m^t) \in \mathbb{R}^{B \times T \times C \times \frac{H}{\eta} \times \frac{W}{\omega}} \quad (6)$$

where $\eta$ and $\omega$ are two hyperparameters. Larger values lead to smaller feature map sizes, which reduce memory consumption but may degrade generation quality. This presents a trade-off issue, requiring a balance between memory efficiency and output quality.

We found that the feature chunk only works at the denoising stage based on the experiments. If the video memory usage of the decoding stage is higher than that of the denoising stage, even if we reduce the memory usage of the denoising stage to 0, the overall peak memory will not change. Previously, we optimized the height and width dimensions of the five dimensions {batch, frames, channels, height, width}. While we can't prune the channel dimension without changing the model architecture, we can consider the batch and frames dimensions:

$$F \in \mathbb{R}^{(B \times T) \times C \times H \times W} \quad (7)$$

In the decoding stage, the decoding function will merge {batch, frames} to $batch \times frames$. At this time, the model has integrated conditional and unconditional information. For one sample, $batch = 1$, so it is equivalent to processing all the frames at once:

$$F \in \mathbb{R}^{T \times C \times H \times W} \quad (8)$$

| Method | Time (s) | Speed Up | LPIPS (↓) | PSNR (↑) | SSIM (↑) | Peak Memory | Peak Reserved |
|---|---|---|---|---|---|---|---|
| **AnimateDiff-Light (U-Net)** 8 steps | | | | | | | |
| Euler | 3.84 | 1.00× | - | - | - | 13562 | 23430 |
| DeepCache (N=2) | 2.87 | 1.34× | 0.164 | 28.64 | 0.7152 | 15200 | 24744 |
| FME | 4.07 | 0.95× | 0.609 | 27.81 | 0.4064 | 4868 | 5688 |
| Our (N=2) | 2.89 | 1.33× | 0.164 | 28.64 | 0.7152 | 5559 | 6238 |
| Our (N=3) | 2.41 | 1.59× | 0.205 | 28.42 | 0.6654 | 5559 | 6238 |
| **SVDI2V-XT (U-Net)** 25 steps | | | | | | | |
| Euler | 47.39 | 1.00× | - | - | - | 15346 | 31084 |
| DeepCache (N=2) | 28.65 | 1.65× | 0.039 | 34.32 | 0.9262 | 21416 | 37570 |
| FME | 57.23 | 0.83× | 0.018 | 37.10 | 0.9649 | 10148 | 11552 |
| Our (N=2) | 30.92 | 1.53× | 0.031 | 34.33 | 0.9271 | 13937 | 15964 |
| Our (N=3) | 24.55 | 1.93× | 0.061 | 33.55 | 0.9189 | 13937 | 15964 |
| Our (N=4) | 21.27 | 2.23× | 0.074 | 32.22 | 0.8837 | 13937 | 15964 |
| Our (N=8) | 16.57 | 2.86× | 0.251 | 28.84 | 0.8345 | 13937 | 15964 |
| **AnimateDiff (U-Net)** 50 steps | | | | | | | |
| Euler | 33.50 | 1.00× | - | - | - | 13564 | 23944 |
| DeepCache (N=2) | 20.67 | 1.62× | 0.127 | 31.75 | 0.9017 | 16843 | 27576 |
| FME | 35.86 | 0.93× | 0.556 | 28.46 | 0.5325 | 5898 | 6688 |
| Our (N=2) | 22.33 | 1.50× | 0.127 | 31.75 | 0.9017 | 8583 | 9908 |
| Our (N=3) | 18.09 | 1.85× | 0.172 | 30.55 | 0.8581 | 8583 | 9908 |
| Our (N=4) | 15.96 | 2.10× | 0.243 | 30.08 | 0.8417 | 8583 | 9908 |
| **TextToVideo (U-Net)** 25 steps | | | | | | | |
| Euler | 5.03 | 1.00× | - | - | - | 7334 | 10814 |
| DeepCache (N=2) | 3.45 | 1.46× | 0.226 | 29.57 | 0.6856 | 8157 | 11592 |
| FME | 7.55 | 0.67× | 0.039 | 35.17 | 0.8956 | 4048 | 4724 |
| Our (N=2) | 4.82 | 1.04× | 0.225 | 29.57 | 0.6842 | 3939 | 4606 |
| **EasyAnimate (DiT)** 25 steps | | | | | | | |
| Euler | 78.63 | 1.00× | - | - | - | 34884.07 | 36830.00 |
| DeepCache (N=2) | - | - | - | - | - | OOM | OOM |
| FME | - | - | - | - | - | - | - |
| Our (N=2) | 59.88 | 1.31× | 0.410 | 28.39 | 0.0743 | 29593.33 | 31596.00 |
| Our (N=3) | 41.14 | 1.91× | 0.548 | 28.43 | 0.0583 | 29593.33 | 31596.00 |
| Our (N=5) | 25.94 | 3.03× | 0.590 | 28.43 | 0.0555 | 29593.33 | 31596.00 |

Table 1: **Result**: DeepCache leads to a surge in memory usage, while FME decreases the speed-up, and both may cause quality degradation across different models. In contrast, our approach not only preserves quality but also accelerates inference and reduces memory consumption to below the baseline model. N means that the feature map is recalculated once every N steps.

Obviously, the amount of GPU memory at this time is extremely large, and the methods we introduced earlier do not involve calculations at this stage. We consider the video as a combination of multiple pictures and use VAE slicing von Platen et al. (2022) to decode the video frame by frame. It should be noted that this method turns a single computation into multiple computations, but under the acceleration of DeepCache, the time consumed by decoding is negligible. Finally, we merge the all the images $\hat{I}_n$ to get the video:

$$\hat{I}_{0:T} = \{VAE(F_n)\}_{n=0}^{T}, F_n \in \mathbb{R}^{C \times H \times W} \tag{9}$$

# 3 EXPERIMENTS

## 3.1 ANALYSIS

We adopted the default Euler scheduler and compared DeepCache, FME, and our method. For prompt, we use the default prompt (Image and Text) in the huggingface von Platen et al. (2022) to generate the video. All experiments were conducted on four NVIDIA L40S GPUs (45GB each) with a batch size of 1. The resolution was set to 512×512 for all methods, except for SVD where 1024×576 was used. The number of inference steps was fixed at 8 for AnimateDiff-Light, 25 for StableVideoDiffusion and TextToVideo, and 50 for AnimateDiff. In addition, the earliest version of DeepCache focused only on accelerating U-Net architecture models. Although Fora claims to have accelerated DiT architecture models, the open-source code is incomplete, and Fora is for image generation, so we extend DeepCache to DiT architecture models (EasyAnimate). However, since

| Method | Time (s) | LPIPS (↓) | PSNR (↑) | SSIM(↑) | Encode Memory | Denoise Memory | Decode Memory |
|---|---|---|---|---|---|---|---|
| **AnimateDiff-Light** 8 steps | | | | | | | |
| DeepCache | 2.88 | 0.164 | 28.64 | 0.7152 | 3829.13 | 6845.11 | **15200.83** |
| Our (N=2) | 2.89 | 0.164 | 28.64 | 0.7152 | 252.46 | **5559.15** | 4183.40 |
| - swapping | 2.57 | 0.164 | 28.64 | 0.7152 | 3829.13 | **6845.11** | 6115.49 |
| - slicing | 3.46 | 0.164 | 28.64 | 0.7152 | 252.46 | 5559.15 | **11547.01** |
| - chunk | 3.34 | 0.164 | 28.64 | 0.7152 | 252.46 | **5559.15** | 4183.40 |
| **SVDI2V-XT** 25 steps | | | | | | | |
| DeepCache | 28.69 | 0.031 | 34.33 | 0.9262 | 5924.08 | 17202.37 | **21416.39** |
| Our (N=2) | 30.92 | 0.031 | 34.33 | 0.9271 | 2987.84 | **13936.91** | 9254.12 |
| - swapping | 30.97 | 0.031 | 34.33 | 0.9272 | 5924.08 | **16639.12** | 11863.77 |
| - slicing | 31.03 | 0.031 | 34.33 | 0.9272 | 2987.84 | 15439.1 | **17269.52** |
| - chunk | 28.77 | 0.031 | 34.33 | 0.9262 | 2987.84 | **16002.1** | 9211.18 |
| **AnimateDiff** 50 steps | | | | | | | |
| DeepCache | 20.57 | 0.127 | 31.75 | 0.9017 | 3827.81 | 9864.46 | **16842.62** |
| Our (N=2) | 22.33 | 0.127 | 31.75 | 0.9017 | 255.45 | **8583.48** | 5835.14 |
| - swapping | 20.57 | 0.127 | 31.75 | 0.9017 | 3827.81 | **9864.46** | 7757.28 |
| - slicing | 22.32 | 0.127 | 31.75 | 0.9017 | 255.45 | 8583.48 | **13198.53** |
| - chunk | 22.21 | 0.127 | 31.75 | 0.9017 | 255.45 | **8583.48** | 5835.14 |
| **TextToVideo** 25 steps | | | | | | | |
| DeepCache | 3.44 | 0.227 | 29.57 | 0.6842 | 3532.90 | 5394.76 | **8157.18** |
| Our (N=2) | 4.82 | 0.225 | 29.57 | 0.6842 | 659.88 | **4279.35** | 3545.69 |
| - swapping | 4.49 | 0.227 | 29.57 | 0.6842 | 3532.90 | **5090.40** | 4522.34 |
| - slicing | 4.08 | 0.227 | 29.57 | 0.6842 | 659.88 | 4279.35 | **4804.40** |
| - chunk | 3.55 | 0.227 | 29.57 | 0.6842 | 659.88 | **4582.85** | 3539.35 |

Table 2: **Abalation Study**: We remove each of the proposed modules and demonstrate the effectiveness of our approach through the memory consumption changes observed across the three stages.

the original model already had excessively high memory consumption, running DeepCache resulted in out-of-memory (we use 4 L40S, 45GB per GPU) issues. Moreover, FME is not applicable to DiT architectures, so no experimental results are available for either method.

In terms of acceleration, DeepCache (N=2) achieves a speed-up increase from 34% to 65% over the baseline, while Ours (N=2) achieves an increase from 4% to 53%. In contrast, FME does not provide acceleration and may even lead to a slowdown. Beyond acceleration, we further evaluate peak memory usage to understand the trade-offs introduced by different methods. The peak memory usage of DeepCache increases the most, by approximately 11% to 40%. Furthermore, as the number of generated frames increases, the peak memory consumption grows proportionally—for example, at 50 frames it reaches 30026MB, an increase of about 96%. FME markedly reduces peak memory by approximately 34% to 64%, with a change in speed-up from -33% to -5%, but leads to a substantial loss in generation quality. Our method provides a more balanced trade-off: maintaining quality (LPIPS, SSIM, and PSNR) close to DeepCache (More cases can be seen in A), while confirming that the acceleration does not come at the expense of perceptual quality and achieving faster speed—slightly lower than DeepCache—but with peak memory usage significantly lower than both the baseline and DeepCache. Due to the presence of cached feature maps, our peak memory is higher than that of FME. Overall, our approach consistently outperforms the baseline, DeepCache, and FME, offering the most favorable balance across speed, quality, and memory (see Table 1).

## 3.2 ABLATION STUDIES

In this section, we conduct ablation studies to evaluate the effectiveness of our proposed method. We first analyze the contribution of each module to peak memory reduction by selectively removing them and measuring their impact across different stages of the pipeline, then investigate whether our approach depends on the choice of sampling scheduler by comparing it with several alternatives.

**Three Modules** To thoroughly investigate the impact of our proposed three modules on the overall peak memory usage, we designed a series of ablation experiments by selectively removing each module to isolate its contribution. Specifically, we first evaluated the baseline case using only Deep-Cache, and we observed that peak memory was mainly dominated by the denoising and decoding stages. Then we implemented our method and found a notable improvement, that the peak memory consumption decreased simultaneously across all three stages, demonstrating the collective effec-

| Method | Time (s) | Speed Up | LPIPS (↓) | PSNR (↑) | SSIM (↑) | Peak Memory | Peak Reserved |
|---|---|---|---|---|---|---|---|
| DDIM | 58.07 | 1.00× | - | - | - | 15789 | 27626 |
| + Ours(N=2) | 37.22 | 1.56× | 0.264 | 31.72 | 0.7843 | 14078 | 18452 |
| PNDM | 60.32 | 1.00× | - | - | - | 15352 | 27626 |
| + Ours(N=2) | 35.50 | 1.699× | 0.505 | 31.13 | 0.6344 | 14133 | 20146 |
| Euler* | 47.39 | 1.00× | - | - | - | 15346 | 31084 |
| + Ours(N=2) | 30.92 | 1.53× | 0.031 | 34.33 | 0.9271 | 13937 | 15964 |

Table 3: Different sampler on SVDI2V-XT, * means this sampler is the official default sampler.

tiveness of the modules in reducing memory demand. To further compare the contribution of each module, we removed them individually and measured peak memory across the three stages. The results show that: 1) Removing swapping significantly increased peak memory in all three stages. Swapping, which is built upon the CPU offloading functionality of the diffusers library, involves transferring both model weights and cached feature maps. Its effect spans all stages. 2) Removing slicing significantly increased peak memory in the decoding stage. This is because VAE slicing processes frames sequentially rather than all at once during decoding. 3) Removing chunking significantly increased peak memory in the denoising stage. This is because only chunking is applied to feature maps specifically during the denoising process (See in Table 2).

**Scheduler**  We consistently adopted the official default sampling scheduler. In the ablation experiments, we also included DDIM and PNDM. The baseline runtime varied slightly across different schedulers, and after applying our method, the quality degradation also differed depending on the scheduler. Among them, the default Euler scheduler achieved the best performance on time cost, LPIPS, PSNR, and SSIM (See in Table 3). These experiments demonstrate that our method is not constrained by the choice of scheduler.

### 3.3 VISUALIZATION

We compared the frame-by-frame quality differences of videos generated by each method against the original model. Among the three metrics, PSNR and SSIM are more reflective of low-level pixel and structural similarity, while LPIPS better aligns with human perception. The results are visualized using violin plots, which show that LightCache maintains consistency with DeepCache, whereas FME produces unstable outcomes (See in Figure 5 and 6).

## 4 RELATED WORKS

### 4.1 TRAINING-FREE ACCELERATION OF DIFFUSION MODELS.

Training-free acceleration methods aim to improve the inference efficiency of diffusion models without modifying their parameters or requiring additional training. These approaches typically exploit structural redundancies or adaptively skip computation steps during the sampling process Ma et al. (2024b); Yu et al. (2023); Zhan et al. (2024); Li et al. (2023a); Tian et al. (2025); Zhou et al. (2025). The reverse process of diffusion models inherently involves the step-by-step reconstruction of images or videos Ho et al. (2020). However, certain intermediate states, such as latent representations and attention maps, contain natural redundancy or sparsity, which provides the foundation for training-free acceleration strategies Ma et al. (2024b). These methods include: 1) Inference-skipping, where redundant steps with minimal change or intermediate activations with large zero regions are omitted to reduce unnecessary computations Ma et al. (2024b); Tian et al. (2025); Yu et al. (2023); Zhou et al. (2025). 2) Computation graph optimization, where inference engines (e.g., ONNX Runtime, TensorRT, TVM) restructure the execution graph to eliminate redundant operations and reduce memory access overhead while preserving mathematical equivalence. 3) Model quantization, where model weights and activations are quantized from high to low precision, reducing memory usage and accelerating computation without retraining Siddegowda et al. (2022); Wang et al. (2025); Zhao et al. (2025; 2024a); Shang et al. (2023). In general, training-free approaches often require a careful trade-off between speed improvement, memory savings, and acceptable levels of accuracy degradation.

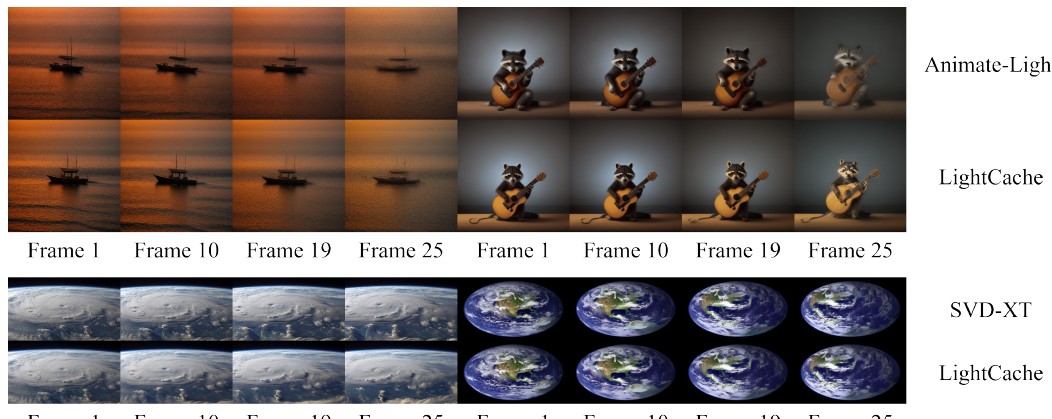

Figure 7: 4 cases for Animate-Light and SVD-XT

## 4.2 CACHE IN DIFFUSION MODELS.

Cache temporarily stores parts of main memory that are expected to be accessed again soon Smith (1982). Recently, a lot of work has applied the cache mechanism in diffusion models Ma et al. (2024b); Agarwal et al. (2024); Wimbauer et al. (2024); Ma et al. (2024a); Zhou et al. (2025); Liu et al. (2025). By carefully analyzing the model's computation workflow, inference efficiency can be significantly improved. One effective strategy is to reuse intermediate or final representations generated during earlier timesteps of computation. This approach avoids redundant recalculations, which not only accelerates the overall inference process but also reduces computational resource consumption and latency. Ma et al. (2024b); Wimbauer et al. (2024); Zhou et al. (2025) leverage the temporal redundancy in the reverse denoising process of diffusion models by caching and reusing features from specific layers, thereby avoiding redundant computations and accelerating the generation process. Agarwal et al. (2024) significantly reduces computational cost and latency by reusing intermediate noise states from the diffusion process, allowing new requests to skip a portion of the denoising steps and condition on cached intermediate states generated from similar previously processed text prompts. FreeDoM leverages energy functions to provide guidance during the diffusion sampling process via energy gradients Yu et al. (2023). EasyCache uses a dynamic threshold to determine when it is safe to reuse cached results Zhou et al. (2025). PAB leverages the U-shaped variation pattern of attention outputs in DiT models during the diffusion process, and reuses the stable-phase attention features through a pyramid-style broadcasting strategy, thereby significantly reducing redundant computations Zhao et al. (2024b). FORA reduces redundant computation by caching and reusing the intermediate features of the Attention and MLP layers at fixed intervals Selvaraju et al. (2024). These advancements demonstrate the growing potential of cache-based strategies to improve the efficiency and scalability of diffusion model inference.

## 5 CONCLUSION

In this work, we proposed **LightCache**, a training-free framework that achieves low-memory and accelerated video generation without compromising output quality. By analyzing the inference process across encoding, denoising, and decoding stages, we introduced three complementary strategies: Asynchronous swapping, Feature chunk, and VAE slicing. Compared to baseline model and DeepCache, and FME, these stage-specific optimizations substantially reduce peak memory usage while keeping the additional time overhead well below the acceleration gains. In future work, we plan to explore combining LightCache with complementary training-free methods, as well as extending it to DiT models, longer video sequences and multi-modal generation tasks where memory efficiency and inference speed are critical.

## 6 ETHICS STATEMENT

This work does not involve human subjects, personally identifiable data, or sensitive information. We believe it does not raise ethical concerns.

## 7 REPRODUCIBILITY STATEMENT

We have made our code available in the anonymous project. Detailed hyperparameters, inference procedures, and data processing steps are provided in the code file. Random seeds are fixed for reproducibility, and all results have been verified across multiple independent runs.

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

# A APPENDIX

## A.1 LLM USAGE STATEMENT

Large language models (LLMs) were employed to assist in refining the clarity, grammar, and style of the manuscript. The use of LLMs was limited to language polishing and did not contribute to the generation of substantive ideas, analyses, or results. All scientific content, interpretations, and conclusions are the authors' own.

## A.2 CASES PRESENTATION

**Text-to-Video** Video.8's prompt is *"masterpiece, best quality, highly detailed, ultra detailed, sunset, orange sky, warm lighting, fishing boats, ocean waves, seagulls, rippling water, wharf, silhouette, serene atmosphere, dusk, evening glow, golden hour, coastal landscape, seaside scenery."*

Video.9's prompt is *"raccoon playing a guitar"*

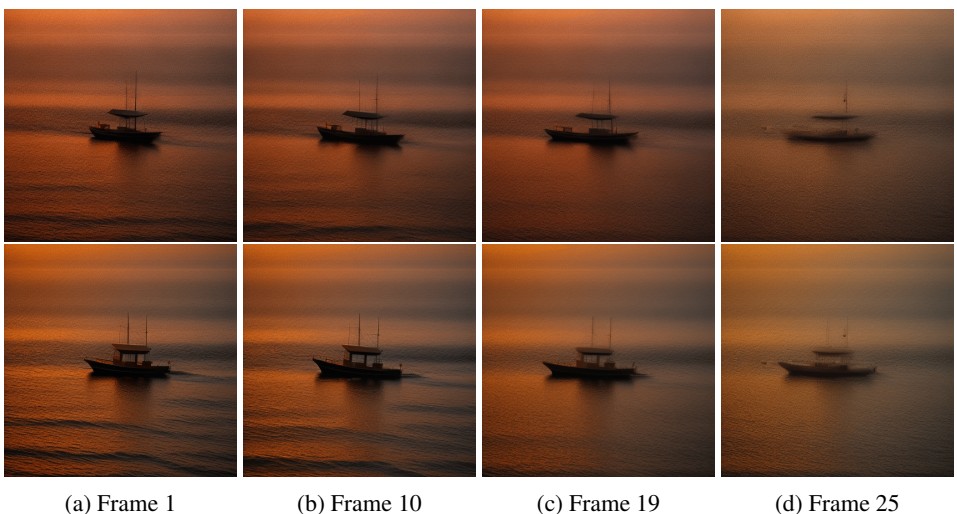

(a) Frame 1      (b) Frame 10      (c) Frame 19      (d) Frame 25

Figure 8: Upper: Original Model. Lower: LightCached Model

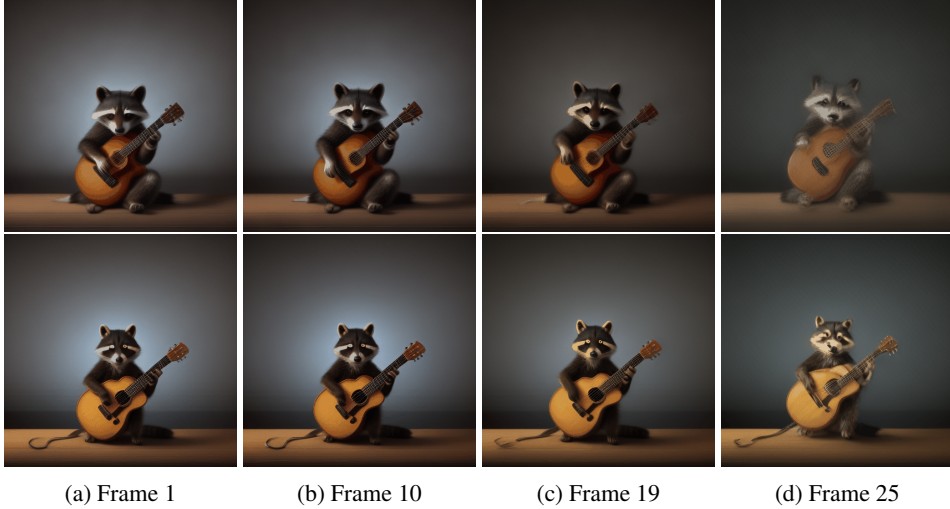

(a) Frame 1      (b) Frame 10      (c) Frame 19      (d) Frame 25

Figure 9: Upper: Original Model. Lower: LightCached Model

**Image-to-Video** The input of the following videos is taken from `https://pixabay.com`.

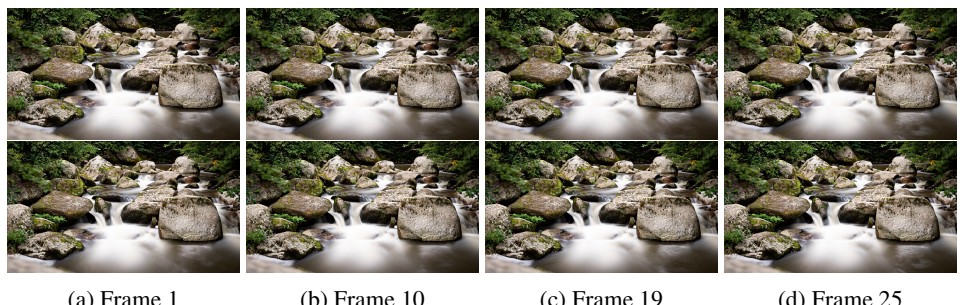

    (a) Frame 1          (b) Frame 10          (c) Frame 19          (d) Frame 25

Figure 10: Upper: Original Model. Lower: LightCached Model

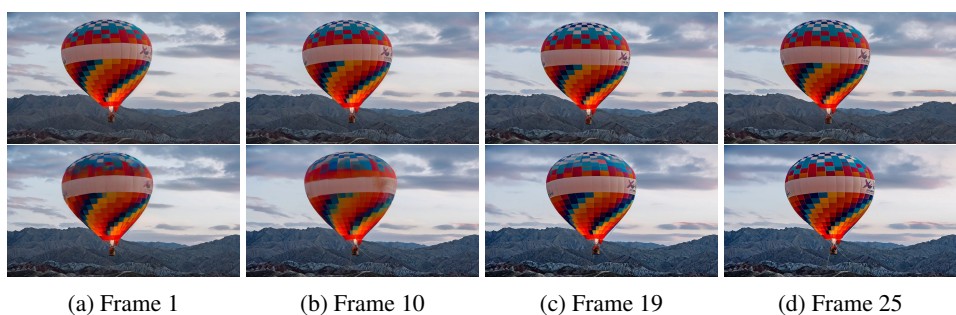

    (a) Frame 1          (b) Frame 10          (c) Frame 19          (d) Frame 25

Figure 11: Upper: Original Model. Lower: LightCached Model

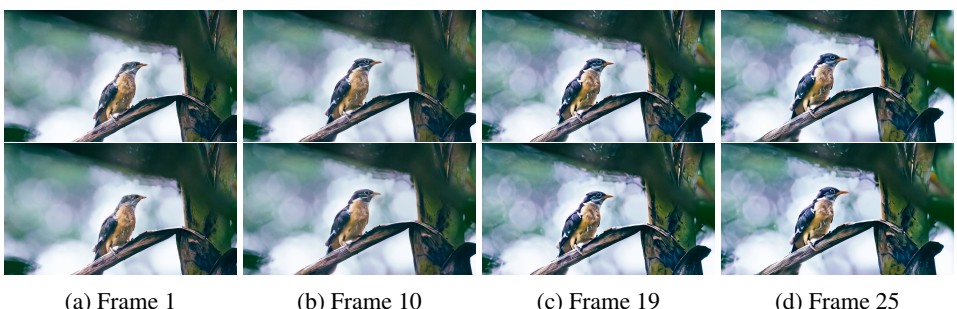

    (a) Frame 1          (b) Frame 10          (c) Frame 19          (d) Frame 25

Figure 12: Upper: Original Model. Lower: LightCached Model

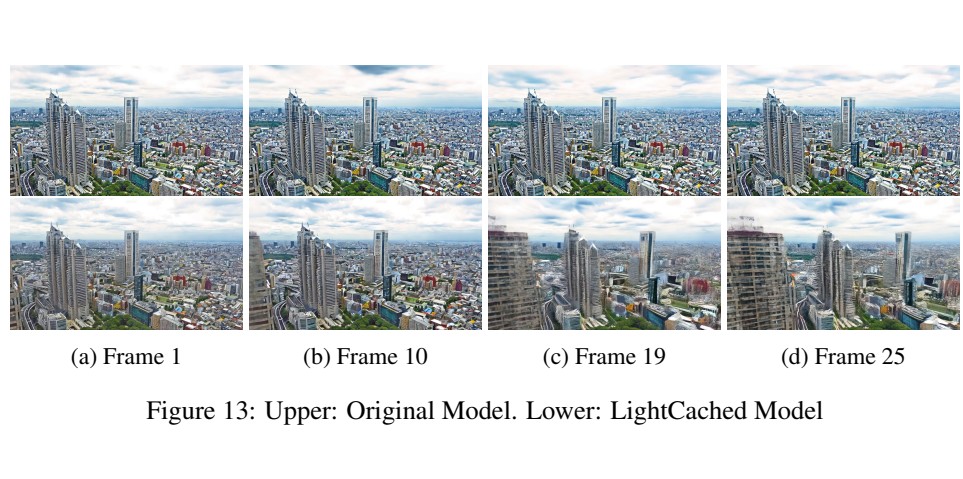

| (a) Frame 1 | (b) Frame 10 | (c) Frame 19 | (d) Frame 25 |

Figure 13: Upper: Original Model. Lower: LightCached Model

| (a) Frame 1 | (b) Frame 10 | (c) Frame 19 | (d) Frame 25 |

Figure 14: Upper: Original Model. Lower: LightCached Model

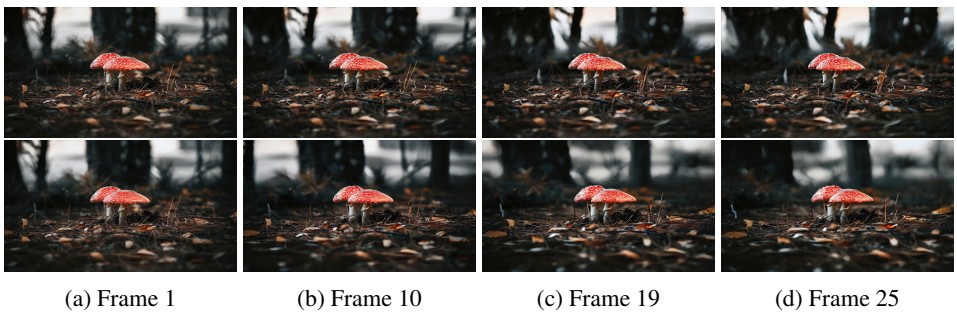

| (a) Frame 1 | (b) Frame 10 | (c) Frame 19 | (d) Frame 25 |

Figure 15: Upper: Original Model. Lower: LightCached Model

