# OpenReview forum: "LightCache: Memory-Efficient, Training-Free Acceleration for Video Generation"
_ICLR.cc/2026/Conference — ICLR 2026 Conference Withdrawn Submission_

### Official Review · Reviewer_mtf3 · 2025-10-26

**Soundness:** 2
**Presentation:** 2
**Contribution:** 2
**Rating:** 4
**Confidence:** 4

**Summary:**

This paper presents LightCache, a training-free framework for accelerating diffusion-based video generation with improved memory efficiency. By introducing Asynchronous Cache Swapping, Feature Chunking, and VAE Slicing, the method mitigates memory surges in denoising and decoding stages, achieving faster inference and lower GPU usage than DeepCache with minimal quality loss.

**Strengths:**

The paper provides a systematic analysis of memory bottlenecks in diffusion-based video generation and proposes simple yet practical strategies to alleviate them without retraining, offering a useful engineering reference for future work on efficient inference.

**Weaknesses:**

1. The writing quality needs improvement, as several parts are confusing. For example, the latter part of the Introduction does not clearly explain how the proposed slicing strategy works, but only describes why it is needed.

2. The discussion and citation of related work are insufficient.

    a) In particular, for the choice of base models, many more recent and memory-intensive DiT-based architectures, such as HunyuanVideo and WAN 2.2, have emerged. It remains unclear whether the proposed method, which is primarily validated on U-Net–based models, can generalize effectively to these DiT architectures or to models with significantly larger parameter scales.

    b) Secondly, cache-based acceleration methods have been extensively explored in video generation. It is recommended that the authors adopt a more advanced method specifically designed for video diffusion, such as TeaCache, for experimental comparison. Moreover, the paper lacks adequate discussion of recent cache-based approaches in this domain.

3. Many of the techniques mentioned in the paper, such as asynchronous CPU offloading, have already been explored in prior engineering practices, and therefore the proposed method lacks sufficient novelty.

4.Typo: Line 201 — “cacheing” should be corrected to “caching.”

**Questions:**

1. It is unclear what specific model the TextToVideo baseline refers to in the paper, as there is no corresponding citation or reference provided.

---

### Official Review · Reviewer_QJjv · 2025-10-30

**Soundness:** 2
**Presentation:** 2
**Contribution:** 2
**Rating:** 2
**Confidence:** 5

**Summary:**

This paper proposes LightCache, a training-free framework that reduces GPU memory during video diffusion inference by asynchronously offloading cached tensors, chunking feature maps, and slicing VAE decoding.

**Strengths:**

+ The paper employs a series of engineering techniques to achieve cache-based acceleration while simultaneously reducing GPU memory consumption.

**Weaknesses:**

- The proposed techniques, such as feature chunking and asynchronous CPU offloading of cached tensors, are widely used engineering practices in existing approaches (e.g., PAB). As a result, the methodological novelty of the paper appears limited, and the contribution may not be strong enough to justify publication in its current form.

- The writing quality could be improved for clarity and precision. Several parts of the paper are difficult to follow. For example, the Abstract claims that cache-based acceleration increases memory consumption in both the denoising and decoding stages. While the denoising overhead is understandable, the paper does not provide sufficient analysis or justification for why cache-based methods would increase memory usage during decoding.

**Questions:**

- There already exist more advanced cache-based acceleration methods for video generation, such as FasterCache [1], TeaCache [2], and PAB [3]. It would be valuable if the authors could include experimental results comparing their method with these approaches, in addition to DeepCache, which was originally designed for image synthesis with UNet architectures.

- The base models selected for evaluation appear relatively limited in scope. More recent and widely adopted architectures such as WAN2.1 and HunyuanVideo exhibit substantially higher memory demands. It would be informative to see how the proposed method performs on these more challenging and representative models.

[1] Lv Z, Si C, Song J, et al. FasterCache: Training-Free Video Diffusion Model Acceleration with High Quality.

[2] Liu F, Zhang S, Wang X, et al. Timestep Embedding Tells: It’s Time to Cache for Video Diffusion Models.

[3] Zhao X, Jin X, Wang K, et al. Real-time Video Generation with Pyramid Attention Broadcast.

**Details Of Ethics Concerns:**

None.

---

### Official Review · Reviewer_iDtT · 2025-10-30

**Soundness:** 3
**Presentation:** 3
**Contribution:** 2
**Rating:** 6
**Confidence:** 3

**Summary:**

This paper explores training-free acceleration for video generation based on diffusion models by addressing memory surges during inference. The authors decompose inference into encoding, denoising, and decoding stages, identifying that cache-based acceleration often increases memory use. To mitigate this, they propose three stage-specific strategies — Asynchronous Cache Swapping, Feature Chunking, and Latent Slicing for Decoding, which reduce memory consumption while keeping the time overhead lower than the acceleration gain. Experiments show that the proposed approach achieves faster inference and lower memory usage with only minor quality degradation compared to the baseline.

**Strengths:**

The authors analyze the problem of increased memory consumption in cache‑based acceleration of video‑generation methods. They propose a training‑free approach specifically for diffusion‑based models, decomposing inference into three stages, encoding, denoising, and decoding, and explicitly addressing the memory overhead associated with the denoising and decoding phases. The paper presents well‑established, yet effective, techniques to mitigate these issues and demonstrates a noticeable improvement in performance. Overall, the work is well written, and evaluated.

**Weaknesses:**

The principal weakness lies in the level of novelty. Most of the contributions already appear in prior work and pertain mainly to implementation‑level details rather than to the fundamental underlying problem. Although the proposed improvements are effective, they contribute little new insight to solving the issue; instead, they address it at an application‑level.

**Questions:**

- How well does the model generalize onto different structural backbones (not only U-Net)?
- In Table 2 we can see that larger steps 25+ impact the time of your method. Is there the possibility of performing a very high step generation to see the overhead from your method?

---

### Official Review · Reviewer_3jhK · 2025-10-31

**Soundness:** 2
**Presentation:** 2
**Contribution:** 2
**Rating:** 2
**Confidence:** 5

**Summary:**

This paper presents LightCache, a training-free framework designed to improve memory efficiency during video diffusion model inference. The authors decompose the process into denoising and decoding stages and propose three corresponding strategies: feature chunking to reduce feature map size, asynchronous swapping to offload cached features to the CPU, and sliced VAE decoding to handle frames sequentially. Experiments across several video generation models demonstrate that these methods effectively lower peak GPU memory usage while maintaining generation quality.

**Strengths:**

1.The paper is easy to follow and conceptually clear.
2.The proposed methods are effective in reducing memory consumption.

**Weaknesses:**

1.There is almost no novelty—chunking, swapping, and slicing are all standard memory optimization techniques already used in many diffusion frameworks.

2.The method shows no real advantage on multiple models (SVD, AnimateDiff, TextToVideo); speed-up at N=2 is minimal, and larger N values severely degrade generation quality.

3.Baseline comparison is too limited, focusing mainly on DeepCache, without strong evidence of improvement.

4.The approach largely repackages existing engineering tricks, trading memory for time rather than offering genuine acceleration.

5.The ablation study is insufficient, with limited analysis of how each proposed component (chunking, swapping, slicing) quantitatively contributes to overall performance.

**Questions:**

How is the trade-off between memory reduction and generation quality quantitatively controlled in the proposed methods? Are there guidelines for choosing the chunking or swapping parameters?

---

### Note · Authors · 2025-12-02

I have read and agree with the venue's withdrawal policy on behalf of myself and my co-authors.